# Exploring Applicability of Different Ecological Protection Measures for Soil and Water Loss Control of Highway Slope in the Permafrost Area: A Case Study of Qinghai-Tibet Highway in China

**DOI:** 10.3390/ijerph20064907

**Published:** 2023-03-10

**Authors:** Xiaochun Qin, Anchen Ni, Dongxiao Yang, Bing Chen, Shiliang Liu

**Affiliations:** 1College of Civil Engineering, Beijing Jiaotong University, Beijing 100044, China; 2China Academy of Transportation Science, Beijing 100013, China; 3School of Environment, Beijing Normal University, Beijing 111875, China

**Keywords:** Qinghai-Tibet Plateau, highway slope, soil erosion, ecological protection, inflow experiment

## Abstract

A variety of slope water and soil conservation measures have been taken along the Qinghai-Tibet Highway, but the systematic comparison of their erosion control ability needs to be strengthened, especially in the permafrost area. To explore the applicability of different measures to control runoff and sediment yield, field scouring experiments were conducted for different ecologically protected slopes, including turfing (strip, block, full), slope covering (gravel, coconut fiber blanket), and comprehensive measures (three-dimensional net seeding). Compared with the bare slope, the bulk density of the plots with the ecological protection measure decreased, the moisture-holding capacity and the organic matter increased correspondingly, and the average runoff velocity also decreased. The soil loss and runoff had a similar trend of different ecological protection measures. The relationship between the cumulative runoff and sediment yield of different measures exhibited a power function, with the increase of scouring flow and the runoff reduction benefit and sediment reduction benefit in different ecological protection-measured plots showing a decreasing trend. The average runoff reduction benefit decreased from 37.06% to 6.34%, and the average sediment reduction benefit decreased from 43.04% to 10.86%. The comprehensive protection measures had the greatest protection efficiency, followed by turfing, while the cover measure had limited improvement. Soil characteristics, vegetation coverage, and the scouring inflow rate are key factors that influence protection efficiency. The results suggest that comprehensive measures and turfing be taken rather than cover measures or bare slopes. This work provides an experimental reference for ecological protection methods for highway slopes in the permafrost area.

## 1. Introduction

Soil erosion is a global environmental problem and generally recognized as the major driver of land degradation worldwide [1,2,3]. It causes not only significant on-site losses, such as fertile topsoil loss [4], a decline in land productivity [5], and loss of cultivated land [6] but also serious off-site impacts, such as river blockage [7] and water body pollution [8]. Therefore, it is very necessary to study the soil and water conservation measures that can limit soil erosion and its impact.

Generally speaking, soil erosion refers to the process by which soil parent material is destroyed, eroded, transported, and deposited under the influence of external forces such as water, wind, freeze–thaw, or gravity [9,10]. Soil erodibility is usually quantified by the K factor in the universal soil loss equation (USLE) and the revised universal soil loss equation (RUSLE) [11]. There are two main methods to quantify K factor. The first is to convert the direct measurement method of K factor based on the long-term monitoring data of standard runoff plots. The disadvantage of this method is that it is time consuming and expensive [12]. The second method is to design models by regression of some existing parameters, including the nomograph model [13], erosion/productivity impact calculator model (EPIC) [14], Torri model [15], and the Shirazi formula [16]. Overall, different methods have been developed to determine soil erodibility, among which empirical models and field measurements are best considered [17,18]. On the road slopes of the Loess Plateau and rainy areas in the south of China, many field-scouring experiments were conducted to investigate the characteristic of soil erosion. Wen et al. [19] analyzed various soil erosion control measures adopted in different periods, and the balance of water and soil conservation was evaluated. Based on the laboratory inflow experiment, Zheng et al. [20] studied the effect of slope gradient on soil erosion under continuous rainfall conditions and derived the critical value of the slope gradient. Qian et al. [21] obtained linear relationships between different combinations of Re, flow velocity, stream power, and sediment yield in a simulated rainfall experiment on eroded slopes. Ji et al. [22] found that the change in soil erosion and sediment yield on the slope was affected by the percentage of soil erosion area. When the percentage of area was greater than 3.3%, the water erosion and sediment yielded significantly increased.

In addition, some research results have shown that soil erodibility is closely related to environmental factors (soil physical and chemical characteristics, vegetation cover and root characteristics, cover type, topography, and geomorphology) [23,24,25,26]. First, the enhancement of soil anti-erodibility is mainly attributed to the improvement in soil structure, soil shear strength, and soil aggregate stability [27,28,29]. Bulk density (BD) and soil texture, among others, have significant effects on aggregate soil stability, infiltration, and soil shear strength [18]. BD can reflect soil pore size and water storage capacity and intervene in soil erosion processes mainly by affecting soil infiltration [30]. Saturated soil hydraulic conductivity (Ks) reflects soil permeability and affects soil erosion to some extent [31]. In addition, the fundamental reason for the reduction of runoff and soil loss is that vegetation alters hydrodynamic mechanisms. Tang et al. [32] found that soil loss was more readily regulated by the fraction and spatial pattern of vegetation cover than runoff. For the rainy areas of southern China, coconut fiber blankets were usually applied to the protection of slope erosion. Coconut fiber blankets and vegetation root systems form horizontal and vertical bidirectional reinforcement effects, which can effectively improve the erosion resistance of the slope, and the plant root system can effectively intercept runoff and sediment [33,34,35,36]. In addition, increasing vegetation coverage is conducive to increasing the roughness of the slope surface, and strengthening the soil and water conservation capacity [37,38]. Similarly, the topographic and geomorphic characteristics can affect the distribution of vegetation cover and land use types, thus affecting soil erodibility and properties [24,25]. Although these findings provided great references for slope soil erosion control, few studies were focused on the plateau area; indeed, some problems may still be encountered due to the high altitude and low temperature. Therefore, it is essential to investigate the protection efficiency of different ecological measures through field experiments.

The Qinghai-Tibet Plateau is a unique natural geographic unit in China. Its unique topography, climate, soil, and biogeochemical process characteristics make the ecological environment of the Qinghai-Tibet Plateau sensitive and fragile. Once damaged, it is difficult to recover [39,40]. With the reconstruction and expansion of the Qinghai-Tibet Highway in recent years, the problem of slope soil erosion requires special attention. During the construction of the highway, a large amount of excavation, backfilling, and earthworks not only caused many exposed slopes but also inevitably affected the surrounding ecological environment, including soil stability [41,42]. Additionally, precipitation and snowmelt converged on the surface to produce runoff. The runoff took away the sediment and caused soil erosion on the slope, finally damaging the ecological environment [43,44,45,46]. For example, Xu et al. [47] used runoff parcels on the exposed slope of the Qinghai-Tibet Highway to investigate the law of runoff and sediment yield on the surface of the slope. They found that runoff and sediment yield decreased with increasing hillslope length, and the annual erosional modulus was as high as 1.19 × 10^4^ t·m^–2^. Shan et al. [48] conducted the scour test by field drainage on the exposed slope of Gonghe-Yushu Highway in Qinghai Province, with the results showing that sediment yield initially increased and then decreased with increasing drainage duration. The Qinghai-Tibet Highway is an untraversed region where the natural condition is relatively harsh. There are many engineering protections along the Qinghai-Tibet Highway slope that have good performance for slope soil erosion in the short term [41], but will cause additional disturbances to the highway slope. The long-term protection effect is difficult to maintain under the influence of the severe climate in the Qinghai-Tibet plateau [42]. Previous studies mainly focused on specific ecological or engineering protection measures yet lack a systematic comparison. In fact, it should be mentioned that various types of ecological protection measures are also used along the Qinghai-Tibet Highway [41,43], including comprehensive protection measures (three-dimensional net seeding), turfing measures (full, strip, block), and covering measures (gravel, coconut fiber blanket). Moreover, excellent water reduction can be obtained by prefabricated grid+grass planting [49] and three-dimensional meshed grass planting measurements [50] on the slope surface of the Qinghai-Tibet Highway. However, the actual performance of soil erosion control and the characteristic of runoff and sediment yield by different ecological protection measures applied to the Qinghai-Tibet Highway slope are not clearly known. Thus, the applicability of different ecological protection measures in the control of water and soil loss of highway slope in permafrost regions needs to be further studied. The appropriate ecological protection measures are of great significance for soil erosion control of the Qinghai-Tibet Highway slope.

In summary, there is a lack of research on soil erosion control of road slopes in the permafrost region of the Qinghai-Tibet Plateau, and the ecological protection measures adopted lack specificity and have poor control of slope erosion; there is also a lack of systematic comparison among various measures. Thus, this study aimed to experimentally quantify soil erosion and surface runoff in the permafrost area of the Qinghai-Tibet Highway. In this study, we observed the runoff and soil loss processes of various slope ecological protection measures through runoff plot tests, analyzed the physical and chemical properties of soil corresponding to different ecological measures, thoroughly discussed the influence of key factors on the protection efficiency, and finally compared the most suitable ecological protection measures and quantified soil and water conservation laws for slopes. It provides an experimental and scientific basis for the slope protection construction of the Qinghai-Tibet Highway.

## 2. Materials and Methods

The research and analysis of the applicability of different ecological protection measures for soil and water loss control of highway slope in the permafrost area entailed the following steps as shown in Figure 1: (1) select the study area and investigate the natural environmental conditions and commonly used ecological protection measures, etc.; (2) design and material preparation of simulated runoff scouring experiment for highway slope; (3) carry out the experiment, measure, and record; and (4) analyze the data under different dimensions.

### 2.1. Experimental Area and Materials

After surveying the site, the existing road slopes of the Permafrost Experimental Station in Madoi County are chosen as the experimental region to avoid the permafrost thaw zone as much as possible, where the natural restoration takes more than three years. Among them, slope selection mainly takes into account slope angle, slope direction, vegetation, season, and precipitation, which are also the main factors affecting freeze–thaw erosion. The six types of ecological protection measures are shown in Figure 2, including gravel cover protection (GCP), three turfing protection layouts (full turfing protection (FTP), strip turfing protection (STP), and block turfing protection (BTP)), three-dimensional net seeding protection (TDNSP) and coconut fiber blankets protection (CFBP). The bare slopes (BS) are taken as control. The experiment region (33°50′4″ N, 99°20′30″ E) belongs to the plateau climate with an annual average temperature of −4.1 °C, and the daily temperature varies greatly. The average annual rainfall was 303.9 mm, which varied greatly from year to year. The precipitation was concentrated from June to September, with rain and heat over the same period. The altitude was about 4500 m, but the terrain was not undulating and relatively flat. The Qinghai-Tibet Highway map is shown in Figure 3.

The soil in the experimental region was mostly alpine meadow soil. The soil layer was very thin and susceptible to erosion. The mechanical composition of the soil on the slope was tested, as shown in Table 1. According to the classification standard (USDA Taxonomy), the soil type of the test plot was sandy loam. The predominant grasses were cyperaceous and gramineous, i.e., Alpine Sagebrush (*Kobresia pygmaea*), Drooping Wildrye (*Elymus natans*), Speargrass (*Splendid achnatherum*), and Alpine Rockjasmine (*Androsace alpine*) [41].

According to the technical code of practice on water and soil conservation monitoring in the water conservancy industry standard of the People’s Republic of China (SL 3-277) [51], inflow simulation experiments were conducted to study the soil erosion control effects of different protection measures. The layout of runoff plots is shown in Figure 4a, and the field experiment was conducted. The gradient of the slope was 33.7°, and there was no other disturbance during the test. Each tested slope was constructed with two 2 m × 5 m runoff plots to ensure that the hydrological conditions were independent and unaffected by each other. The plot was surrounded by a nickel-clad steel sheet with a thickness of 1 mm. The height of the nickel-clad steel sheet was 50 cm (30 cm under the ground and 20 cm above the ground). The collecting tank was installed at the bottom of the plot to collect the runoff samples. The inflow setup is shown in Figure 4b.

### 2.2. Experimental Design

The calculation formula for the flow level design is:(1)$$F_{d}=\frac{W_{d}}{D}\cdot I\cdot A\cdot \cos G\cdot a$$
where *F_d_* is the discharge flow rate, L/min; *W_d_* is the design shallow trench width; *D* is the shallow trench spacing; *I* is the rainfall intensity; *A* is the catchment area; *G* is the gradient of slope; and *a* is runoff coefficient and is set to 0.9. Through calculation, the discharge flow rate is within the range of 2.25–20.5 L/min. The final design discharge flow rate *F_d_* is 2.5, 5, 10, 15, and 20 L/min. (This data is determined based on the single-wide flow caused by heavy rain in the area and the former preliminary experiment results) [52]. The water supply device is closed after 20 min of scouring, and the runoff time is recorded.

Water was supplied by a sprinkler truck, and a flow control valve was installed to ensure a stable head and flow during the experiment. A water tank with 100 cm in length, 20 cm in width, and 10 cm in depth was set at the top of the runoff plot. The tank was placed horizontally and parallel to the slope of the runoff plot to ensure that the initial water that flowed into the runoff plot was uniform and consistent after the water first passed through the water tank and then entered the slope surface. Before the test, the discharge flow rate was determined, first in the runoff plot, geotextile was then laid on the slope to allow the water flow to enter the collecting tank, and a standard bucket was used to take water samples 3 to 5 times. To ensure the accuracy of the flow rate, the error between the control flow and the designed discharge flow was within ±5%. Once the flow rate had met the requirements, the experiment was run, each trial was repeated twice, and the timer was run when the water in the tank was slowly and smoothly discharged down a shallow slope. When the runoff reached the collecting tank, the time of runoff production was recorded. 

### 2.3. Experimental Measurements Methods

After the runoff occurred on the slope, the record started and numbered standard runoff barrels were put at the outlet of the collecting tank for runoff sediment samples. The samples were collected every 1 min in the first 3 min, and the time for each sample was 1 min; after 3 min, samples were taken every 3 min, and the time for each sample was 3 min. The design flow time was 20 min until the end of the test.

Three observation sections were set on the runoff plot at 1.0, 3.0, and 5.0 m from the top, and there were 3 measuring slope sections, ranging from 0–1 m, 143 m, and 3–5 m. After the runoff flowed on the slope, the KMnO_4_ dye tracer method was used to measure the runoff velocities every 5 min. Finally, different velocity correction coefficients were selected to calculate the average velocity according to the flow regime. At the same time, each slope was measured with a thin steel ruler with an accuracy of 1 mm. The width of the cross-section of the water flow and the runoff depth of each slope section were measured with a precision needle of 0.2 mm.

After the test, according to the order of sample collection, the sediment samples in the bucket were fully stirred to make the sediment particles disperse evenly. After the stirring, a numbered and known sampling bottle was used to take samples quickly in the bucket. Sampling was repeated 3 times to ensure that the sediment samples in the bucket had been stirred uniformly. After sampling, the volume of runoff sediment in the sampling bottle was measured, and the electronic scale was used to measure the quality of the runoff sediment samples. The sediment samples were dried at 105 °C, and the quality of the sediment was weighed with an electronic scale with an accuracy of 0.01 g. The sediment concentration was calculated, and the sediment yield per unit area was obtained by unit conversion. Taking the yield from the bare slope as the control group, the effectiveness of runoff and sediment reduction on the slopes of different measures were calculated as:(2)$$EI=\frac{Y_{b}-Y_{m}}{Y_{b}}\times 100\%$$
where EI is an index of effectiveness; $Y_{b}$ is the runoff yield (or sediment yield) of the bare slope; and $Y_{m}$ is the runoff yield (or sediment yield) of the slope with protection.

After the experiment, the shape of the erosion gully was observed, and the measurement indicators included the width and depth of the erosion gully [53].

The measurement of soil physical properties was mainly performed by collecting soil samples using the cutting-ring method. Samples were collected according to the location of runoff plots, and sampling points were arranged at the upper, middle, and lower parts of each plot, with the sampling depth as 0–20 cm; 3 samples were collected with a total of 21 soil samples. The laser particle size distribution meter was used to determine the mechanical composition of the soil. The potassium dichromate external heating method was used to determine the total soil organic carbon content [54,55,56]. The vegetation coverage was measured by the sampling method. First, part of the small cells was identified, and then a sample needle was vertically laid down in the vegetation at the node of each cell. The percentage of the number of sample needles contacting the plant branches and leaves in the total number of sample needles was the vegetation coverage.

### 2.4. Data Analysis

Statistical analysis and figures were performed using the Origin (OriginLab Inc., Washington, DC, USA). Analysis of variance (ANOVA) was conducted to examine significant differences between the runoff, runoff velocity, and sediment concentration among the treatments of the seven measures. The values presented in this study were the averages of the two repeated treatments. For multiple comparisons, the least significant difference (LSD) method was used at the 95% confidence level [42]. To determine the equations of fit for cumulative sediment yield and runoff yield, a nonlinear fitting method was applied using MATLAB R2018a software (MathWorks Inc., Natick, MA, USA). Maps of the experimental plots and the inflow setup were created using AutoCAD 2014 software (Autodesk Inc., San Rafael, CA, USA). The remaining data were analyzed using Microsoft Excel 2019 software (Microsoft Inc., Redmond, WA, USA).

## 3. Results and Discussion

### 3.1. Soil Physical and Chemical Properties

To investigate the soil physical and chemical properties of different ecological measured slopes, the soil was collected from different plots. The soil properties are shown in Table 2.

#### 3.1.1. Soil Bulk Density

The soil bulk density of the entire tested area fluctuates from 1.06–1.49 g/cm^3^. The soil bulk density from small to large is as follows: TDNSP < STP < BTP < FTP < CFBP < GCP < BP. Note that soil bulk density fluctuates slightly. The bulk density of TDNSP is the smallest, while the BP is the largest. In fact, different slope protection measures will affect surface temperature, humidity, and wind speed, thus changing the soil microenvironment and microtopography and forming corresponding soil characteristics. The smallest soil bulk density in TDNSP may be caused by the fiber material of the three-dimensional net, which results in loose and porous soil and good vegetation growth. The three-dimensional net allowed the plant roots to cross and promoted growth evenly. Three-dimensional net seeding also held the gauze pad, turf and soil surface together. Then, a solid composite protective layer was formed [42]. The bare slope is built without artificial grass, the soil is relatively compact due to natural settlement, and the root system of the vegetation is small, so the soil bulk density is the largest.

#### 3.1.2. Soil Capillary Water Holding Capacity and Saturated Water Holding Capacity

It can be seen from Table 2 that there was fluctuation in the soil capillary moisture capacity and saturation moisture capacity of each plot. For TDNSP, both the moisture-holding capacities were the largest, reaching 35.84% and 42.46%, respectively, followed by STP, whose capacities were 30.36% and 34.82%, respectively. The BP soil had the smallest water storage capacity, and its capacities were 20.13% and 21.48%. respectively. The difference is due to the high vegetation coverage and the plant roots interspersed in the community after the use of vegetative measures, which increases the soil porosity to a certain extent. It should be mentioned that the void left by the death of the plant root and the biological activities around the root system has increased the noncapillary pores of the soil, which promoted the virtuous circle of the soil structure, the formation of good soil permeability, and the facilitation of the infiltration and accumulation of water [57,58,59].

#### 3.1.3. Soil Organic Matter

The analysis of soil organic matter content is helpful to understand the water and soil conservation performance of soil. In general, the higher the soil organic matter content is, the stronger the soil’s moisture-holding capacity will be [60]. As can be seen from Table 2, the soil organic matter content in the plots where ecological measures are implemented is higher than in the bare plots, with an increase of 9.52–80.95% unit volume. The content is significantly higher in the TDNSP and STP than in others, which may also be related to large vegetation coverage.

### 3.2. Runoff and Soil Loss Processes

#### 3.2.1. Analysis of Initial Runoff Time

The initial runoff time reflects the influence of soil erosion on the slope surface under different protection measures. As shown in Figure 5, the initial runoff time was extended greatly in the plots that had been protected, especially in TDNSP and turfing plots, where the extension range was 9–70%. But with the increase of the inflow rate, the time of all plots showed a decreasing trend. Nevertheless, the TDNSP still had the longest initial runoff time while BP had the shortest initial runoff time. It should be noted that the difference in initial runoff time among all measures became smaller.

The reason is that the runoff first infiltrated after the scouring water entered the slope. When the flow was small, the runoff velocity was slow too, so that the runoff could stay on the slope for longer and the infiltration time was longer; the runoff was formed after the soil layer was wet. As the flow increased, the infiltration time was shortened, and the initial runoff time was correspondingly advanced. While the soil became saturated, the excess water formed the slope runoff [59]. On the other hand, the results also showed that the vegetation coverage affected the hydrodynamic characteristics of the slope flow to a certain extent and further affected the slope erosion [42,61].

#### 3.2.2. Runoff Processes

The magnitude of runoff on highway slopes is one of the leading factors affecting soil erosion, and the characteristic of runoff can reflect the effect of different ecological protection measures on slope water-holding capacity. The runoff rate of each ecological protection measure under different inflow rate is shown in Figure 6. The runoff rate showed a similar rend under different measures. The runoff rate under the same cross-section and the same inflow rate increased with the extension of time and finally stabilized, which was consistent with the findings of previous studies [9]. The reason for this outcome is as follows: after the runoff started, the soil infiltration rate decreased with the increase of soil water content, leading to the runoff rate increasing and then stabilizing. Compared with BP, the others’ runoff rate was significantly reduced. Under the 2.5–5 L/min inflow rate, the average stable runoff rate was as follows: TDNSP < FTP < STP < GCP < BTP < CFBP < BP; under the inflow rate of 10–20 L/min, the performance was as follows: TDNSP < STP < BTP < FTP < GCP < CFBP < BP. Overall, it seems that TDNSP had the best water-holding capacity, and among the three turfing measures, the FTP effectively slowed the runoff rate. In addition, the runoff rate of GCP and CFBP were both notably higher than other ecological protection measures.

The main reason for different runoff velocities under different measures is that the influence of the increase in the percentage of vegetation cover effectively promoted its roughness. The roughness of the slope surface increased the turbulence and swirl of the runoff in the local area, which made the runoff stay on the slope surface for a longer time and thereby increased the water infiltration [38]. In general, different ecological protection measures all have the effect of reducing runoff velocity, then effectively increasing soil infiltration, and finally holding water.

#### 3.2.3. Soil Loss Processes

It can be seen in Figure 7 that with the increase in inflow duration, the sediment concentration in different protection measures had a similar trend, the sediment concentration first increased rapidly to the extreme value, then gradually decreased, and finally stabilized. When the inflow rate was 2.5 L/min, the sediment concentration of the BP was much higher than that of other protection measures, and the fluctuation of the sediment concentration during the soil loss processes under the other protection measures was small. This indicated that the ecological protection measure adopted had different degrees of interception protection on the slope when the flow was small. As the scouring flow increased, the sediment concentration of GCP increased greatly. When the scouring flow increased to 15 L/min at that time, the average runoff sediment concentration of GCP was not significantly different from that of BP (*p* < 0.05). Overall, when the inflow rate was small, ecological protection measures effectively intercepted the sediment. It should be noted that TDNSP had a relatively stable soil-fixing ability, followed by FTP and STP.

This phenomenon can be explained by the fact that the main infiltration occurs at the beginning of the experiment, the erosion flow is converted into limited runoff whose carrying capacity is relatively small, so the erosion rate is low. As the water inflow continues, the soil moisture content increases, the soil infiltration rate decreases, the scouring flow is converted into more runoff, and the runoff carrying capacity is gradually promoted, aggravating the soil erosion. After the erosion develops to a certain extent, the soil is saturated, the surface runoff is stable, and the erosion rate is reduced to a more stable state. Secondly, the runoff first chooses to transport fine particles, as the inflow time increases, the fine particle content gradually decreases. During the later erosion period, the large aggregates are dispersed by the stripping effect of runoff erosivity. When the large aggregates are stripped away, a relatively large amount of sediment will be lost with the runoff. Obviously, the ecological protection measure can effectively reduce the soil loss on the highway slope, especially the TDNSP and turfing measures.

#### 3.2.4. Relationship between Cumulative Runoff and Soil Loss

Figure 8 and Figure 9 show that under different scouring flows, the cumulative runoff yield increased slowly with the increase of time, and then increased steadily at a fixed rate. Accumulated sediment concentration showed an exponential trend, and with the passage of time, the rate of increase accelerated, indicating that the longer the scouring time, the stronger the sediment production capacity of different scour streams.

The function of the cumulative runoff yield and the cumulative sediment yield for all inflow rate was built, and it was found that the relationship between them satisfies the function y = Ax^B^. The dependent variable y is the cumulative sediment yield, the variable x is the cumulative runoff yield, A and B are the coefficients, and the fitting relationship is shown in Table 3. The goodness of fit is above 0.98. According to the basic concept of runoff and sediment production on the slope, coefficient A is defined as the benchmark for sediment production, which characterizes the ability of sediment production at different flow rates, depending on the soil properties of the slope. The definition of coefficient B is the rate of sediment production, which depends on the magnitude of runoff erosivity. From Table 3, it can be seen that A and B of TDNSP are the smallest, indicating it had the best sediment interception capacity, followed by STP and FTP. This fitting function can act as a reference for soil erosion control for highway slope, especially in the permafrost region.

### 3.3. Analysis of Protection Efficiency of Different Measures

#### 3.3.1. Protection Efficiency of Different Measures

The runoff reduction benefits (RRB) and sediment reduction benefits (SRB) of each slope can be calculated according to Equation (2), respectively. As can be seen from Figure 10, with the increase of the scouring flow, the RRB and SRB in different ecological protection measured plots had a decreasing trend. The average RRB decreased from 37.06% to 6.34%, and the average SRB decreased from 43.04% to 10.86%. There are differences in the RRB and SRB of each protection measure under different inflow rates, ranging from 10% to 35.77%, and the average SRB was as follows: TDNSP > FTP > STP > BTP > GCP > CFBP. In addition, the average RRB was in descending order as: TDNSP > STP > FTP > BTP > GCP > CFBP, in which TDNSP and turfing measures had significantly higher RRB and SRB. When the scouring flow of GCP reached 15–20 L·min^−1^, its corresponding benefits of sediment reduction were −4.11% and −2.51%, respectively. Negative values appeared because when the flow was excessively large, the runoff was formed directly on the surface. The erosion caused by large runoff and high velocity directly exceeded the erosion reduction capacity of the measure, so it had a negative effect of reducing sediment.

In summary, the use of ecological protection measures effectively reduced soil erosion on slopes. TDNSP and turfing measures were relatively effective in reducing runoff and sediment.

#### 3.3.2. The Influence of Soil Property and Vegetation Coverage

In general, big bulk density of soil indicates that the soil is heavy and has poor air permeability, which is not conducive for water infiltration and is easy to form surface runoff, washing away the surface soil and exacerbating soil erosion. In contrast, small bulk density has the opposite effect. The soil water infiltration and air permeability are directly affected by soil porosity. It is also an important factor that determines the soil’s ability to hold water, which, in turn, is determined by noncapillary porosity, which is the main channel for plants to absorb soil moisture and surface water evaporation. Saturated water-holding capacity and capillary water-holding capacity gradually decrease in response to the degree of intensification of soil erosion. The reason is that intensified soil erosion gradually reduces organic matter, as well as fine particles in the soil, destroying the soil structure. The soil organic matter promotes the formation of soil aggregates, which weakens the looseness and dispersion of the soil. At the same time, it can also increase soil porosity. To comprehensively analyze the impact of soil characteristics on sediment and runoff yield on a slope, correlation analysis was conducted in terms of soil bulk density, capillary capacity, saturated water capacity, water content, soil organic matter content, and vegetation coverage. From Table 4, it can be seen that the runoff yield and sediment yield were significantly negatively correlated with soil capillary water-holding capacity, saturated water-holding capacity, organic matter content, and vegetation coverage. Conversely, they were significantly positively correlated with soil bulk density and soil water content. The soil properties directly affected the strength of soil erosion resistance, and with the increase of vegetation coverage, the cumulative sediment yield on the slope gradually decreased. The raindrop disturbance on the slope flows became smaller with the increase of vegetation coverage, the flow regime was more stable, and turbulence action on the soil was much less; thus, soil erosion became smaller, including the runoff and sediment yield.

#### 3.3.3. The Influence of the Scouring Inflow Rate

Generally, the total runoff increased with an increase in the inflow rate and decreased with an increase in the vegetation cover percentage [42]. In this study, inflow simulation experiments focusing on soil erosion on expressway embankment sideslopes at five inflow rates (2.5, 5, 10, 15, 20 L/min) were conducted to investigate soil erosion processes and analyze control effectiveness.

It was found in Figure 11a that the GCP had the steepest slope followed by BP, indicating that the cumulative sediment yield was greatly affected by the inflow rate. The highway slope where GCP was adopted was even worse than BP measured. Additionally, the STP had the minimum slope followed by TDNSP, meaning that the TDNSP and turfing reduced slope sediment effectively. All measures approximately followed the linear relationship between cumulative sediment yield and inflow rate. In addition, the difference in the cumulative sediment yield between BP and the measured slope increased with the increase of the inflow rate, except GCP.

Figure 11b shows the inflow rate had a positive correlation with the runoff under different measures, indicating that with the increase of water discharge flow, the runoff yield increased. The BP had the steepest slope, while the TDNSP had the minimum slope. The measured slope reduced runoff effectively compared to the bare slope. Furthermore, the performance of ecological protection measures was better than cover protection measures from the slope of the fitting function.

## 4. Conclusions

To find out the suitable ecological protection measures applied in the Qinghai-Tibet highway slope, field scouring experiments were conducted to investigate the protection efficiency of ecological protection measures. Additionally, soil loss processes were observed to summarize the runoff and sediment production laws. Finally, key influence factors were discussed. The main conclusions are as follows:(1)Under ecological protection measures, soil properties are improved, bulk density is reduced, and the water-holding capacity and organic matter content are increased, benefitting soil-infiltration capacity and the development of vegetation cover percentage.(2)The sediment interception capacity of sideslopes is enhanced with the increase in vegetation coverage. Comprehensive measures (three-dimensional net seeding) are most effective at reducing runoff and sediment, followed by turfing (strip, full, block). The effect of cover measures on soil and water conservation is very limited.(3)There is a power function relationship between cumulative runoff and sediment production. The baseline coefficient of sediment production and the coefficient of sediment production speed can reflect the effectiveness of ecological protection measures. Protection efficiency is significantly positively correlated with vegetation coverage and significantly negatively correlated with the scouring inflow rate.(4)It is recommended that comprehensive protection and turfing be used in soil erosion control of the Qinghai-Tibet highway slope and that cover measures (gravel, coconut fiber blanket) and bare slopes be avoided. This study provides an experimental reference for the soil erosion control of highway slopes on the Qinghai-Tibet Plateau and other similar permafrost areas.

## Figures and Tables

**Figure 1 ijerph-20-04907-f001:**
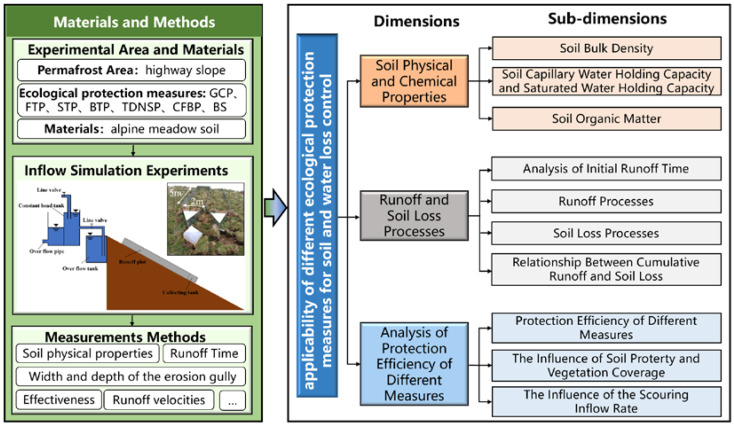
Research framework adopted in this paper.

**Figure 2 ijerph-20-04907-f002:**
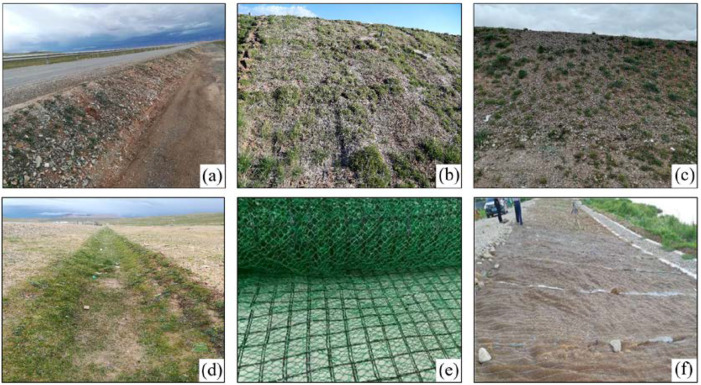
Ecological protection measures. (**a**) Gravel cover protection. (**b**) Full turfing protection. (**c**) Strip turfing protection. (**d**) Block turfing protection. (**e**)Three-dimensional net seeding protection. (**f**) Coconut fiber blankets protection.

**Figure 3 ijerph-20-04907-f003:**
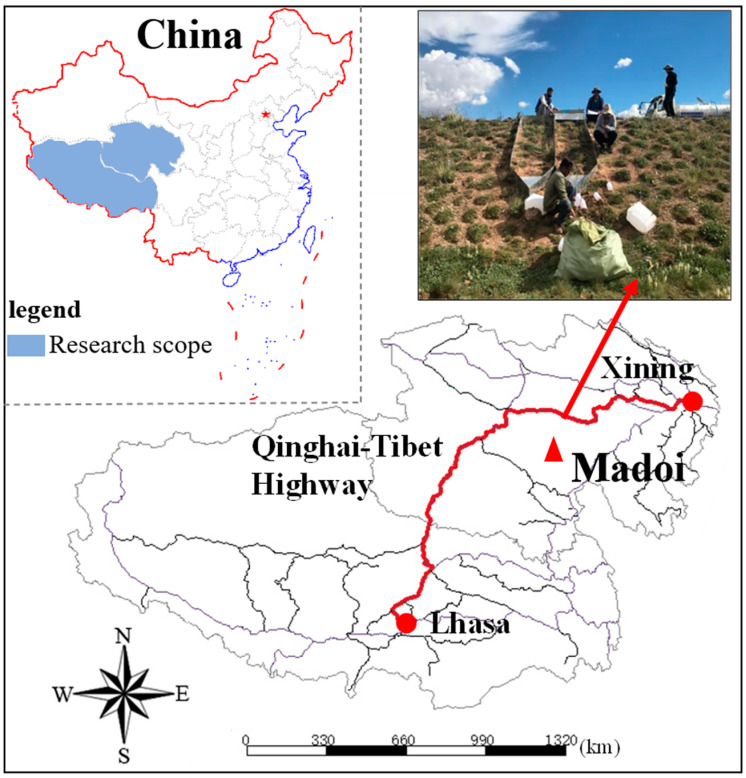
Location of experimental region.

**Figure 4 ijerph-20-04907-f004:**
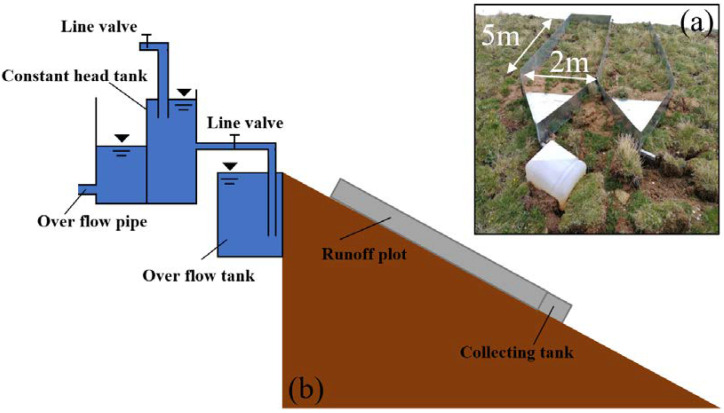
Runoff plots and experimental inflow setup. (**a**) The layout of runoff plots. (**b**) The inflow setup.

**Figure 5 ijerph-20-04907-f005:**
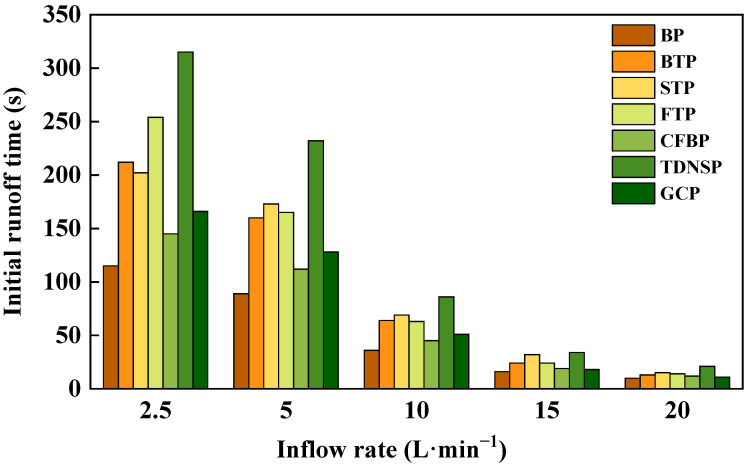
Initial runoff time.

**Figure 6 ijerph-20-04907-f006:**
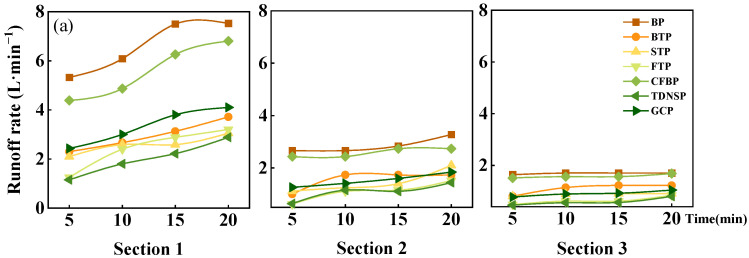
Runoff rates of seven measures under different inflow time. (**a**) Inflow rate = 2.5 L·min^−1^. (**b**) Inflow rate = 5 L·min^−1^. (**c**) Inflow rate = 10 L·min^−1^. (**d**) Inflow rate = 15 L·min^−1^. (**e**) Inflow rate = 20 L·min^−1^. Note: In Sections 1–3, the distance from the observation section to the top of the runoff plot is 1 m, 3 m, and 5 m, respectively.

**Figure 7 ijerph-20-04907-f007:**
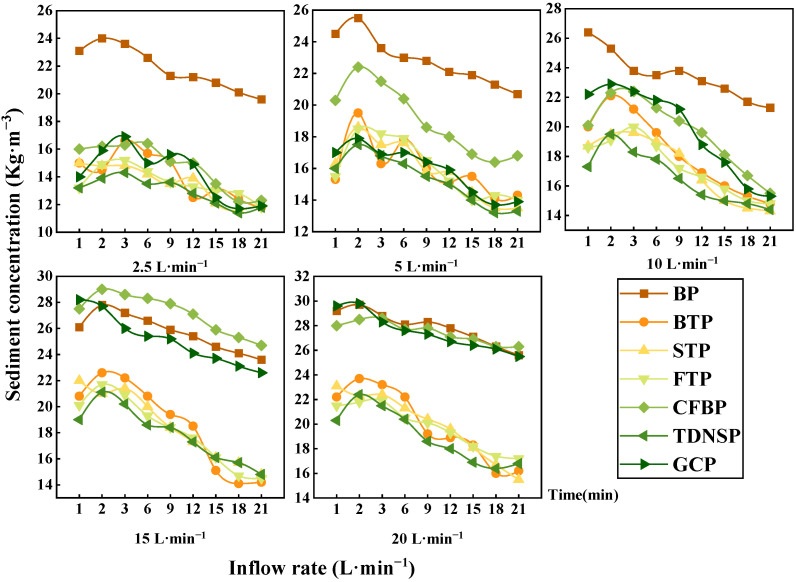
Sediment concentration of seven measures under different inflow times.

**Figure 8 ijerph-20-04907-f008:**
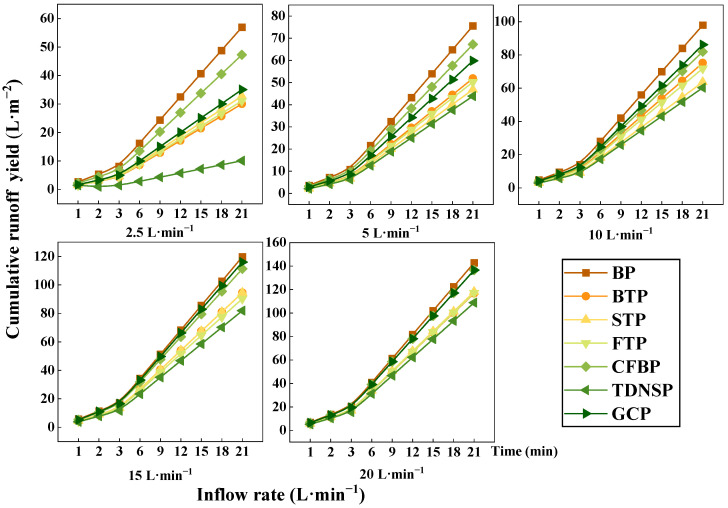
Cumulative runoff of seven measures under different inflow times.

**Figure 9 ijerph-20-04907-f009:**
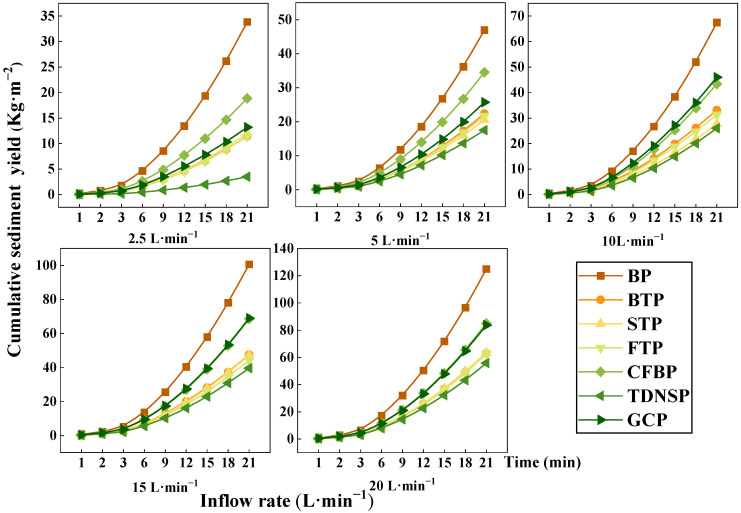
Soil loss of seven measures under different inflow times.

**Figure 10 ijerph-20-04907-f010:**
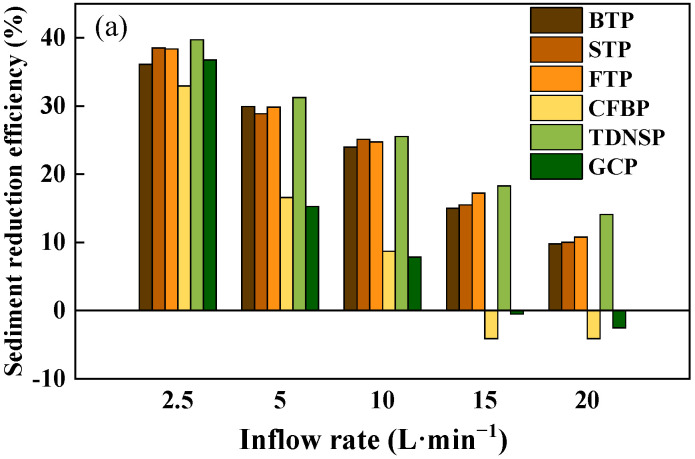
Effectives of seven measures on highway slope. (**a**) RRB of different measures. (**b**) SRB of different measures.

**Figure 11 ijerph-20-04907-f011:**
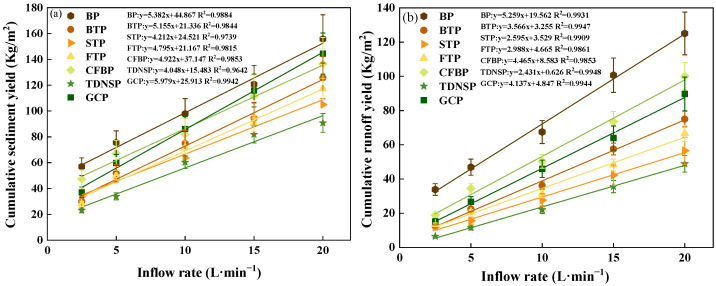
The influence of scouring inflow rate. (**a**) Relationship between inflow rate and cumulative sediment yield. (**b**) Relationship between inflow rate and cumulative runoff yield.

**Table 1 ijerph-20-04907-t001:** Soil mechanical composition under different slope ecological protection measures.

Measures	Clay (<0.002 mm)	Silt (0.02–0.002 mm)	Sand (0.02–2 mm)
BP	9.33%	13.67%	77.00%
BTP	5.67%	14.33%	80.00%
STP	6.33%	12.67%	81.00%
FTP	6.14%	13.59%	79.62%
CFBP	7.33%	14.00%	78.67%
TDNSP	4.00%	26.00%	70.00%
GCP	9.33%	10.67%	80.00%

**Table 2 ijerph-20-04907-t002:** Soil properties of seven measured plots.

Soil Properties	BP	BTP	STP	FTP	CFBP	TDNSP	GCP
Bulk density (g/cm^3^)	1.49	1.2	1.12	1.23	1.26	1.06	1.3
Saturated moisture (%)	21.48	30.02	34.82	28.80	28.45	42.46	26.15
Capillary moisture (%)	20.13	27.51	30.36	26.42	26.82	35.84	23.85
Organic matter content (g/kg)	2.1	3.1	3.8	2.7	2.3	3.7	1.7
Vegetation coverage (%)	20	60	60	65	20	70	10

Note: GCP is gravel cover protection; FTP is three turfing protection layout (full turfing protection); STP is strip turfing protection; BTP is block turfing protection; TDNSP is three-dimensional net seeding protection; and CFBP is coconut fiber blankets protection.

**Table 3 ijerph-20-04907-t003:** Relationship between cumulative runoff and sediment yield under different ecological protection measures.

Measures	Function	Degree of Fitting
BP	y = 0.15x^1.67^	0.98
BTP	y = 0.07x^1.49^	0.99
STP	y = 0.06x^1.51^	0.99
FTP	y = 0.05x^1.43^	0.99
CFBP	y = 0.10x^1.49^	0.98
TDNSP	y = 0.04x^1.1^	0.98
GCP	y = 0.09x^1.55^	0.99

**Table 4 ijerph-20-04907-t004:** Correlation analysis of soil characteristics, vegetation coverage, and runoff and sediment yield under different ecological protection measures.

Soil Property	Runoff Production (L)	Sediment Production (g)
Bulk density (g/Kg)	0.515 **	0.479 *
Capillary water-holding capacity (%)	−0.555 **	−0.483 *
Saturated water-holding capacity (%)	−0.632 **	−0.524 **
Organic matter (g/Kg)	−0.666 **	−0.637 **
Water content (%)	0.626 **	0.414 *
Vegetation coverage (%)	−0.669 **	−0.628 **

Note: * means *p* < 0.05; ** means *p* < 0.01.

## Data Availability

Not applicable.

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
