# Peer review of "Exploring Applicability of Different Ecological Protection Measures for Soil and Water Loss Control of Highway Slope in the Permafrost Area: A Case Study of Qinghai-Tibet Highway in China"

_ijerph, 2023, doi:10.3390/ijerph20064907_

Round 1

Reviewer 1 Report

The title and the work are meaningful, however, it is not published, and some major revision should be conducted. Specific revisions are suggested below.

1, Water erosion is commonly used, whereas hydraulic erosion seems not correct.

2, the sentence “Whether the applied measures… confirmed” should be rewritten.

3, You described seven measures in line 32, however, there were only four photos showing four measures?

4, In Table 1, the words were wrongly used, i.e., powder and Grit should be “Silt” and ”Sand”, respectively.

5, The constructed plots with steel sheet can greatly disturb the soil surface that can greatly influence runoff and sediment production, how do you avoid it? How total plots you built? Fourteen plot in total? You said there were seven measures, and two replications for each one. Again, you said, subsequent tests in line 181, what does it mean? This information should be given.

6, For Figure 4, I cannot understand the usefulness of the left part showing the “Over flop pipe and Constant head tank”. Furthermore, how does the flow running into the plot?I can not see a outlet of the “Over flow tank”.

7, I cannot understand  equation 1. Why did you use Wd and D, I also understand the meaning of “D”.

8, The sentence “unit yield was obtained by unit conversion” in line 213 should be rewritten.

9, The method to obtain “saturated moisture”, “capillary moisture”, and “vegetation coverage “ in Table 2 should be given.

10, The full names of the abbreviations in table 2 should be indicated at the end of the table.

11, The result sections should not contain analyses, JUST results in these sections.

12, The unit of bulk density after 1.49 in line 247 is wrongly given.

13, For Figure 6, what are the sections 1, 2, and 3? They should be given at the end of the Figure caption.

14, The subfigures in Fig. 8 are not consistent to those in Fig.7, why are there sections 1, 2, and 3 in Fig. 8?

15, I am confused for your method. You conducted a flush experiment for the slopes. However, in 4.3., you gave the “influence of rainfall intensity”, it is a rainfall simulation experiment!

Reviewer 2 Report

 In order to explore the applicability of different ecological protection measures to control runoff and sediment yield, the field scouring experiments were conducted .The experiment is designed appropriately, the methods are adequately described, and the results are clearly presented. The article still needs to be improved, as follows:

(1)Both engineering protection measures and ecological protection measures are important. The author should not think that ecological protection measures are more important because he wants to study ecological protection measures. The author needs to make scientific and objective statements.

(2)The research framework adopted in this paper is the author's research ideas, should be stated in the method section.

(3)Authors should pay attention to the use of normative scientific terms, such as soil mechanical composition, and the correct ones are clay, silt and sand.

Round 2

Reviewer 1 Report

I carefully checked the responses to the comments I gave. They have been well revised. This is a hard work and meaningful. I think it can be accepted in its current form.

Author Response

On behalf of my co-authors, we would like to express our sincere thanks to the reviewers for giving us some constructive comments and detailed suggestions on our manuscript entitled "Exploring applicability of different ecological protection measures for soil and water loss control of highway slope in the permafrost area: A case study of Qinghai-Tibet Highway in China" (ID: ijerph-2223880).